# Pinch Grip per SE Is Not an Occupational Risk Factor for the Musculoskeletal System: An Experimental Study on Field

**DOI:** 10.3390/ijerph19158975

**Published:** 2022-07-23

**Authors:** Emma Sala, Nicola Francesco Lopomo, Francesco Romagnoli, Cesare Tomasi, Jacopo Fostinelli, Giuseppe De Palma

**Affiliations:** 1Unit of Occupational Health, Hygiene, Toxicology and Occupational Prevention, University Hospital Spedali Civili, 25123 Brescia, Italy; giuseppe.depalma@unibs.it; 2Department of Information Engineering, University of Brescia, 25123 Brescia, Italy; nicola.lopomo@unibs.it; 3Unit of Occupational Health and Industrial Hygiene, Department of Medical and Surgical Specialties, Radiological Sciences and Public Health, University of Brescia, Piazzale Spedali Civili 1, 25121 Brescia, Italy; francesco.handtherapy@gmail.com (F.R.); cesare.tomasi@unibs.it (C.T.); fostinelli.medlavoro@gmail.com (J.F.)

**Keywords:** pinch, finger grip, strain, EMG, musculoskeletal overload risk assessment, upper limb

## Abstract

Introduction: Some ergonomic evaluation methods define pinch grip as a risk factor independent of the exerted force. The present experimental study was performed with the main aim of objectively measuring the muscle engagement during the execution of pinch grip. Methods: the participants of the study were healthy workers occupationally involved in a high-intensity repetitive job related to the sorting of letters and small packages. Surface electromyography (sEMG) was used to study the activity of the abductor pollicis brevis and first dorsal interosseous fibers related to the execution of the required working tasks, while the force exerted during voluntary muscle contraction for pinch grip was measured by a portable acquisition system. The subjects were specifically asked to exert the maximum voluntary isometric contraction (MVIC) and further voluntary isometric contractions with a spontaneous force (SF) equal to 10%,20% and 50% of the MVIC; finally, the workers were asked to hold in pinch grip two types of envelopes, weighing 100 g and 500 g, respectively. Results: The force required to pinch 100 and 500 g envelopes by the fifteen subjects of the study corresponded to 4 and 5% MVIC, respectively. The corresponding sEMG average rectified values (ARV) were approximately 6% of that at MVIC for first dorsal interosseus (FDI) fibers and approximately 20–25% of MVIC for abductor pollicis brevis (ABP) fibers. Bivariate correlation analysis showed significant relationships between force at MVIC and FDI ARV at MCV. Conclusions: The obtained results demonstrate that muscle recruitment during pinch grip varies as a function of the SF: not only the position but also the exerted force should be considered when assessing the pinch grip as risk factor for biomechanical overload of the upper limb.

## 1. Introduction

Work-related musculoskeletal disorders (WRMSDs) and, among these, upper-limb WMSDs (UL-WMSDs) represent the most common work-related disorders in industrialized countries [1,2] and are among the most prevalent causes of lost-time injuries and illnesses [3]. Traditionally, repetitive industrial tasks have been associated with UL-WMSDs [4]. In Europe, this kind of disorder accounts for approximately 41% of all occupational diseases [5]. The etiopathology of UL-WMSDs is complex since they are multifactorial disorders, exacerbated by high levels of exposure (in terms of intensity, frequency, and/or duration) and both biomechanical and psychosocial risk factors [6]. Among the biomechanical factors, a strong connection was described between hand-wrist disorders and the combination of repetitiveness, strength, awkward postures (i.e., extreme joint positions), and localized vibrations [2,7].

To date, many methodological approaches are available and can be used to assess the risk of developing UL-WMSDs. Focusing on the hand, the evaluation of pinch grip strength represents one of the main aspects to consider when assessing the risk of biomechanical overload in the upper limb [3]. In the human hand, the thumb is indeed of paramount importance since it acts as a stabilizer, while the long fingers realize dexterous activities [8]; it represents what most differentiates humans from other primates, and its role in tool making separates us from the remainder of the animal kingdom [9]. Among thumb joints, the basal joint of the thumb, i.e., the carpometacarpal (CMC) joint, represents one of the most common sites of osteoarthritis and repetitive stress injuries due to both the complexity of its motion and the concentration of mechanical forces [10]. In fact, since pinch grip is ubiquitous at work as well as in everyday life, all joints of the involved fingers but especially the basal joint continue to experience stress [11], especially during isometric functional tasks [12]. Furthermore, the carpometacarpal (CMC) joint of the thumb is the second most common joint in the hand affected by osteoarthritis (OA). This condition that affects the activity of the hands in all aspects of life, besides being painful, produces a significant disability [13]. The forces localized at the tip of the thumb during pinching are transmitted to the basal joint, with an intensity increased from 10 to 13 times. Furthermore, the repetitive development of forces—even in low weight load or isometric conditions—could lead to fatigue of the muscular structures [14]; this information can be of paramount importance to correctly identify task-related ergonomic risk factors. However, it is important to identify force levels in digital gripping below which the risk is not present. Definition and significance of term pinch and related force exerted are still open outside the ergonomic field of interest, too. The lack of consent appears relevant in upper-limb biomechanical risk assessment. Therefore, for biomechanical overload risk assessment, it is important to be able to objectively measure force in pinch and the resulting muscle activation for different force levels. Observational methods for ergonomic risk assessment do not provide a precise definition of strain. In this picture, the myoelectric manifestation of muscle fatigue [15,16] has been receiving growing attention as a potential exposure metric in research toward the prevention of UL-WMSDs [17].

The main aim of the present study was to objectively evaluate the muscle activity involved in repetitive occupational pinch grip used to sort letters and parcels in a selected working population. We evaluated both the exerted force by a dedicated load cell and muscular activity by sEMG surface electromyography. The assessment of muscle activation can allow the determination of the limit of force engagement beyond which occupational overload of the hand becomes a risk factor.

## 2. Materials and Methods

### 2.1. Population and Study Design

This was an experimental cross-sectional study on workers of an international expedition-logistic company involved in high-intensity repetitive work addressing small letter and parcel sorting. No a priori inclusion criteria were fixed. Concerning their work, the subjects, while maintaining a standing position, must sort the envelopes that arrive in boxes from their left, lift them with one hand by using a pinch grip, and put them in special cataloged binders; during such a task, the workers also must scan the items by using a special device that is located on the index and middle fingers of the other hand. The envelope sorting activity is carried out in a work shift of 210 min, divided into 105 min of actual repetitive work and the other 105 dedicated to preparing the workstation, configuring the software, and supplying the consumables. A mean of 1000 envelopes per worker are sorted during each 105-minute shift phase. The task is performed only during the night shift. The workers were enrolled at the end of the work shift to study the worst situation in terms of muscular fatigue. The study was performed in the context of compulsory occupational health surveillance for risk assessment purposes. However, all participating subjects expressed their informed consent. The study followed the ethical principles of the Helsinki Declaration.

### 2.2. Equipment and Experimental Conditions

A 4-channel sEMG apparatus (EMG QUATTRO; OT Bioelettronica, Italy) was used to acquire (with a 1024 Hz sampling rate and 10 Hz–500 Hz analog passband filter) muscle activity during isometric contractions and to estimate the myoelectric manifestation of muscular activity related to the execution of the required working tasks. After skin preparation, concentric pre-gelled bipolar adhesive surface electrodes (CoDe 1.0 B, OT Bioelettronica, Italy) prepared with additional electroconductive gel were placed on the abductor brevis pollicis (ABP) and first dorsal interosseous (FDI) fibers. The sEMG electrodes were placed to acquire the activity of both the ABP and FDI muscles. To correctly place the electrodes, we identified and isolated the defined muscles by following the manual muscle strength testing (MMST) method, as proposed by Bradsma et al. [18]. The MMST is strictly dependent on the examiner’s ability to assess the pressure as a parameter for strength [19]. To avoid this bias, the test was performed by the same assessor.

The force exerted during voluntary muscle contraction for pinch grip was measured by a portable acquisition system (P-forceMet; OT Bioelettronica, Turin, Italy). The analog signal from the load cell, amplified and conditioned in frequency (with a passband analog filter from 0 to 100 Hz), was digitally converted and graphically displayed on a liquid crystal screen to provide visual biofeedback and allow the user to pursue a predetermined target.

The multichannel sEMG apparatus was synchronized with the force acquisition system, a preprocess filtering and denoising procedure was performed to clean the signals, and the data acquired during all the measurements were saved on a Secure Digital (SD)-type memory.

### 2.3. Experimental Steps

The pinch grip force assessment was divided into three phases. In the first phase, the subjects were specifically asked to exert the maximum voluntary isometric contraction (MVIC) with the load cell interposed between the thumb and index finger (Figure 1a), telling subjects to “squeeze the load cell between your fingers with the maximum force you can exert”. During the second phase, the subject repeated the voluntary isometric contractions applying a spontaneous force equal to 10%, 20%, and 50% of the MVIC and maintained them for 1 min each; the visual feedback provided by the system was used to control the force. To obtain this measurement, we asked the subjects to squeeze the load cell to 10% of the MCV by telling them to “squeeze until keeping the force level below the line you see on the monitor that represents 10% of your maximum contraction”. With the same visual feedback on the monitor, we explained to the worker how to apply a force on the load cell equivalent to 20% and 50% of the MCIV.

In the final phase, the workers were then asked to keep two kinds of envelopes, weighing 100 g and 500 g, raised in a pinch grip with the load cell placed between the envelope and the thumb to record the force (Figure 1b); during the test, the force was exerted with the forearm resting on a table, thus isolating only the functional muscular work required to pinch the envelopes.

Then, the sEMG electrodes were placed to acquire the activity of both the ABP and FDI muscles. 

### 2.4. sEMG Data Analysis

Starting from the acquired sEMG raw data, we analyzed the average rectified value (ARV) to assess the fatigue of both ABP and FDI during the tests.

### 2.5. Statistical Analysis

The statistical analysis was performed using IBM SPSS^®^ software version 26.0.1 (IBM SPSS Inc. Chicago, IL, USA) and Stata^®^ software release 16.0 (Stata Corporation, College Station, TX, USA). Variables were distributed normally according to the Kolmogorov–Smirnov test; hence, continuous variables are presented as the means (M) ± standard deviations (SD) and min/max. Correlations between variables were evaluated by Pearson’s test. A two-sided alpha level of 0.05 was used for all tests.

The authors had full access to and take full responsibility for the integrity of the data.

## 3. Results

The study sample was represented by fifteen healthy subjects (mean age 39.1 years, mean length of service 6.9 years) and included all the workers in that company involved in sorting. Table 1 summarizes the main characteristics of the worker sample. 

Table 2 shows the main results of the descriptive analysis of the investigated items for the 15 subjects. A certain concordance was apparent between the measured force and ARVs at the different experimental steps, including the execution of the specific job tasks. On average, the force required to pinch 100 and 500 g envelopes is 4 and 5% of that required at MVIC, respectively. The corresponding ARV values are approximately 6% of MVIC for FDI and approximately 20–25% of MVIC for ABP.

Bivariate parametric correlation analysis showed significant relationships between force at MVIC and FDI ARV at MVIC (r = 0.85, *p* < 0.0001) and force at 20% of MVIC and FDI ARVs SF for 100 and 500 g (r = 0.66, *p* < 0.05; r = 0.73, *p* < 0.05, respectively).

Figure 2 shows the positive linear regression between FDI ARV (dependent variable) values and force at MVIC as independent variable (R² = 0.71, *p* < 0.001); R-squared, the coefficient of determination, measures the strength of the linear relationship between the independent variables X in the regression model and the dependent variable Y. The equation of the regression line is also shown in the box of Figure 2.

No sex differences in ARV values or in force were observed, and no relationship was apparent between ARV and age, with the exception of ABP ARV at MVIC (*p* = 0.0468 for 100% MVIC).

## 4. Discussion

New instruments for the risk assessment of biomechanical overload reported in the recent scientific literature are sEMG and systems of objective measurement of the force in grip or pinch grip [20,21,22,23]. The main aim of such studies is to evaluate objective tools for biomechanical overload risk assessment but also looking for new limit values for strain exertion [24] and evaluation of differences in the mean pinch strength values in terms of age and sex [25].

The main aim of the present study was objectively measuring the muscle engagement in a group of workers engaged in a repetitive pinch grip task. In particular, the study describes the activation of the muscles involved in the pinch grip depending on the applied force at the fingertip.

The obtained results showed a difference in force exerted and muscle activation depending on the workers’ use demand; electrical muscle activation increases simultaneously with the increase in SF as a percentage of MVIC and with the increase in the load supported by the distal phalanges. According to the results, we can state that below a certain force commitment (in our case, the 100 g and 500 g envelopes), the measured ARV is considerably lower than that measured at 10% MVIC. This finding is in agreement with both literature data [26] and our previous experiences [27], which stated that for pinch values lower than 10% of the MVIC, there is no biomechanical overload.

The limits of this study are the small sample and the “on field” parameter recording made on the work environment and not in a laboratory standard condition. As it concerns the first issue, to the best of our knowledge, there are not similar studies in literature useful as reference. As it regards the second issue, it presents the advantage of evaluating a real occupational situation in the field instead of a “simulation” of it. The enrolled workers were recruited at the end of their night work-shift, thus representing the worst scenario in terms of potential muscular fatigue.

This study could represent an important contribution to the research of methods for the assessment of biomechanical risk and the objectification of risk factors. The study represents the first experimental phase of objective risk assessment activities that can be carried out by applying load cells and sEMG to workers during the whole work shift. 

The definition and significance of the term pinch and related force exerted are still open outside the ergonomic field of interest. The lack of consensus appears relevant in upper-limb biomechanical risk assessment. Actions with pinch are numerous in working activities.

Some risk assessment methods (such as OCRA index [28] and ISO 11228/3 [29]) consider action in pinch “at risk” only because they require a posture of opposition of the first two fingers, but others (such as Washington method [30], strain index [31], OREGE [32,33], HAL ACGIH [34]) consider pinch grip “at risk” only when associated with force engagement.

Without objective measurement of muscle engagement while performing the pinch grip, the true biomechanical overload may not be properly considered. In addition, we believe that our results may give a contribution to the comprehension of the osteoarthritis (OA) of the carpometacarpal joint of the thumb, that is, the second most common joint in the hand affected by (OA). This condition that affects the activity of the hands in all aspects of life, besides being painful, produces a significant disability [13].

The limitations of this study are represented by the on-field research and the small sample. Furthermore, as it is one of the first studies to analyze these two muscles, there are few data in the literature that can allow us to check with our results.

## 5. Conclusions

Pinch actions may be numerous in working activities. Some observational ergonomic evaluation methods define pinch grip as a risk factor independent of the exerted force. 

Our results show that muscle recruitment during pinch grip varies as a function of the spontaneous force and that, in the evaluation of UL-WMSDs, the association between risk factors (in our case, the distal phalanges opposition posture and force commitment) is essential to assess the actual extent of the risk: not only the position but also the exerted force should be considered when assessing the pinch grip as a risk factor for biomechanical overload of the upper limb.

## Figures and Tables

**Figure 1 ijerph-19-08975-f001:**
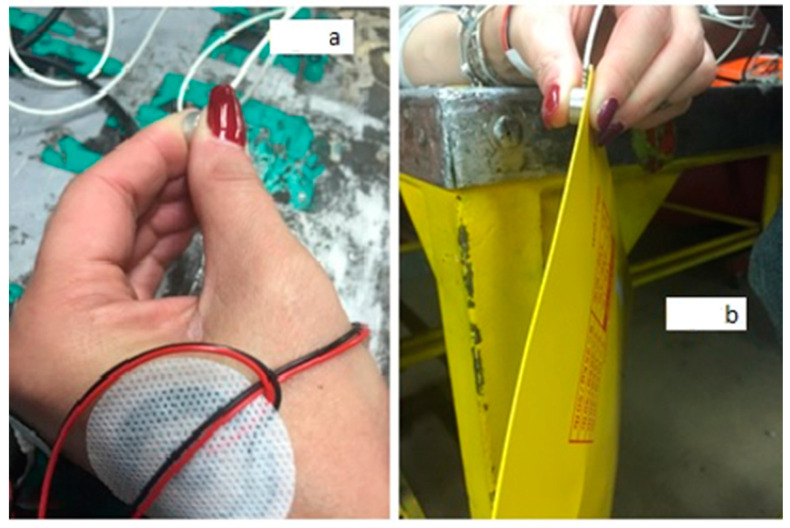
(**a**) Setup used for sEMG and force exerted during MVIC and spontaneous force (SF) in isometric conditions; the force during pinch grip is recorded by using a load cell and a dedicated acquisition system (P-forceMet, OT Bioelettronica, Italy). (**b**) Setup for the acquisition of the sEMG and SF during the lifting of 100 g and 500 g envelopes.

**Figure 2 ijerph-19-08975-f002:**
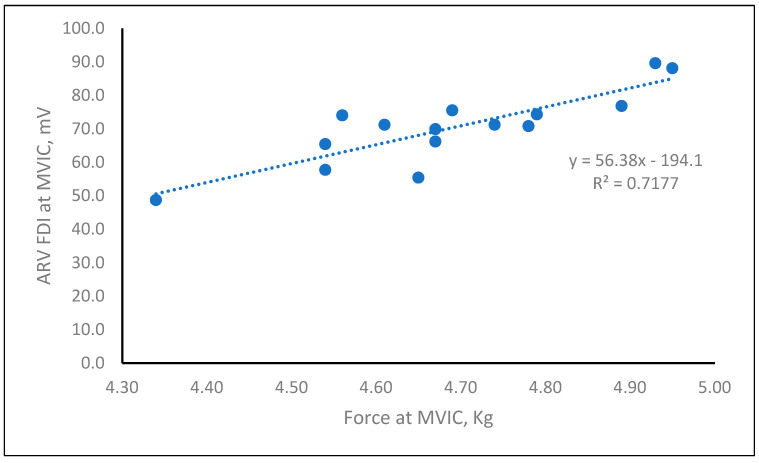
Regression between force and ARV FDI values at MVIC in the enrolled workers.

**Table 1 ijerph-19-08975-t001:** Characteristics of the subjects variables.

	Males (N = 5)			Females (N = 10)		
	Mean ± S.D.	Min.	Max.	Mean ± S.D.	Min.	Max.
Age (y)	37.8 ± 1.6	36	40	39.8 ± 2.6	36	45
BMI	24.3 ± 2.5	20.8	26.9	23.5 ± 3.1	19.8	28.8
Work seniority (y)	6.6 ± 0.6	6.0	7.0	7.1 ± 0.6	6.0	8.0

**Table 2 ijerph-19-08975-t002:** Distribution of ARV values for the first dorsal interosseous (FDI) and the abductor brevis pollicis (ABP) measured during the different experimental steps in the worker sample (N = 15).

Experimental Steps	Force (kg)		FDI ARV (mV)		ABP ARV (mV)	
	M ± SD	% f MVIC	M ± SD	% MVIC	M ± SD	% MVIC
MVIC	4.69 ± 0.17	-	70.32 ± 10.95	-	80.68 ± 7.77	-
10%	0.48 ± 0.09	10	23.08 ± 3.44	33	14.70 ± 1.49	18
20%	1.07 ± 0.10	23	43.77 ± 2.87	62	36.99 ± 2.32	46
50%	2.63 ± 0.34	56	60.45 ± 3.19	86	50.11 ± 2.38	62
Envelope 100 g	0.19 ± 0.10	4	3.84 ± 0.78	6	16.38 ± 2.49	20
Envelope 500 g	0.25 ± 0.11	5	4.04 ± 0.64	6	20.29 ± 2.18	25

## Data Availability

The data that support the findings of this study are available from the corresponding author, A.M., upon reasonable request.

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
