# Peer review of "Pinch Grip per SE Is Not an Occupational Risk Factor for the Musculoskeletal System: An Experimental Study on Field"

_ijerph, 2022, doi:10.3390/ijerph19158975_

Round 1
Reviewer 1 Report
Restructure the abstract as an introduction, methods, results and conclusions. For readers it is not necessary to describe MVC. I recommend the inclusion of the design .. of the participants ..
Furthermore, it should be emphasized that "The carpometacarpal (CMC) joint of the thumb is the second most common joint in the hand affected by osteoarthritis (OA) . This condition that affects the activity of the hands in all aspects of life, besides being painful, produces a significant disability "as a reference: https://doi.org/10.1016/j.apmr.2020.06.012
Sort letters and parcels?
In the design as in the title, I suggest using the term "proof-of-principle", in the methods the results of your selection should not be presented, but the eligibility of your population. So these workers were included, but did they have to work for a number of years? Were only males selected? An age “cap”? Concomitant pathologies?
L95 how was this fatigue calculated? Was there a cutoff?
L115 2 probes? Did you follow any positioning guidelines?
The photo is not very illustrative, the studio design is very interesting why don't you try to draw and describe the various steps you have determined?
L138 move above when describing electromyography
L157 ARV cannot represent fatigue. Is there any reference in the literature? Figures 2 and 3 are superfluous, the readers of the newspaper have the tools to know how to process an MVC, maybe write a few lines of paragraph in a discursive way.
L175 Enter how many subjects you included here
I don't understand the need for figure 4, why not correlate the strength to the MVC of the two muscles examined?
The beginning of the discussion, by convention, should re-propose the objective of the studies and subsequently the major findings of the study.
In this regard, the design is interesting for the relationship between force and strength obtained from the surface EMG, but at this point the title should be changed because I don't understand the risk factor where it was highlighted? What is pinch grip without force?
Furthermore, the term fatigue should be avoided because it is a complex problem and has not been objectively assessed, you could add some scale or survey score of disability, functionality .. otherwise it becomes difficult to accept conclusions drawn in this condition.
Reviewer 2 Report
The article is very interesting, because it studies one of the main incidences problem in the professional environment and the authors use an adequate biomechanical perspective, using EMG for the assessment, this give more interest to the study.
Although the study only has 15 subjects, as the authors themselves describe in the study, it is a first approximation to be able to carry out a larger study and has the strength that it has been carried out in the same workplace as the subjects, although they develop it as a limitation, it can be a strength as it does not take the subject out of the environment where the action under study is carried out.
I only recommend reviewing bibliographical references 30, 33 and 35 as the authors are in capital letters.
Reviewer 3 Report
The authors study the relationship between force and muscle fatigue when performing a pinch grip. The study was well performed and the paper is well written. Some comments might help to further improve the paper.
1) Methods, line 130 and 131: you write:
During the second phase, the subject repeated the voluntary isometric contractions applying a spontaneous force equal to 10%, 20% and 50% of the MVIC and maintained them for 1 minute each;
Can you please explain how it was assessed that the force was 10%, 20% and so on?
2) Conclusion. You write: The definition and significance of the term pinch and related force exerted are still open outside the ergonomic field of interest. The lack of consent appears relevant in upper limb biomechanical risk assessment. Actions with pinch are numerous in working activities and if defined "at risk" only because they require posture of opposition of the first two fingers (as required by some risk assessment methods such as OCRA index [30] and ISO 11228/3 238 [31] but not by others such as Washington method [32], strain index [33], OREGE [34,35], HAL ACGIH [36], we could not classify correctly to the real biomechanical overload.
I propose to move this part to the discussion and maybe you could explain the assessment tools in a bit more detail. The second sentence is very long.
3) I propose to down tune the conclusion a bit. I think questionnaire based assessment tools should be accompanied sophisticated measurements as you used. But they should be add and not replaced.
4) You looked into force. Can you also discuss repetition?
5) Abstract last sentence: I am not sure I understand the following sentence: “the association between risk factors (in our case, the distal phalanges opposition posture and force commitment) is essential to assess the actual extent of the risk”. I think what you want to say is that not only the position but also the exerted force should be considered when assessing the pinch grip as risk factor for pain. Is this correct? Please reconsider the wording.
Thank you for the chance to read this interesting paper.
Round 2
Reviewer 1 Report
25 do not enter the results (15) in the methods ..
65 Furthermore, it should be emphasized that "The carpometacarpal (CMC) joint of the thumb is the second most common joint in the hand affected by osteoarthritis (OA). This condition that affects the activity of the hands in all aspects of life, besides being painful, produces a significant disability "as a reference: https://doi.org/10.1016/j.apmr.2020.06.012
170 Figures 2 and 3 are superfluous, the readers of the journal have the skills to know how to process an MVC, maybe write a few lines of paragraph in a discursive way.
243 limitations should be the last paragraph of the discussion. It lacks a more stringent eligibility, it is the first study to analyze these two muscles, there is no control.
